# Research

evolution, developmental biology, microbiology

evo-devo, social evolution, aggregative multicellularity, morphological evolution, microbial development

**Author for correspondence:**
Marco La Fortezza
e-mail: marco.lafortezza@env.ethz.ch

# Social selection within aggregative multicellular development drives morphological evolution

Marco La Fortezza and Gregory J. Velicer

Institute for Integrative Biology, ETH Zürich, Zürich 8092, Switzerland

MLF, 0000-0002-4485-9840; GJV, 0000-0001-8721-1502

Aggregative multicellular development is a social process involving complex forms of cooperation among unicellular organisms. In some aggregative systems, development culminates in the construction of spore-packed fruiting bodies and often unfolds within genetically and behaviourally diverse conspecific cellular environments. Here, we use the bacterium *Myxococcus xanthus* to test whether the character of the cellular environment during aggregative development shapes its morphological evolution. We manipulated the cellular composition of *Myxococcus* development in an experiment in which evolving populations initiated from a single ancestor repeatedly co-developed with one of several non-evolving partners—a cooperator, three cheaters and three antagonists. Fruiting body morphology was found to diversify not only as a function of partner genotype but more broadly as a function of partner social character, with antagonistic partners selecting for greater fruiting body formation than cheaters or the cooperator. Yet even small degrees of genetic divergence between distinct cheater partners sufficed to drive treatment-level morphological divergence. Co-developmental partners also determined the magnitude and dynamics of stochastic morphological diversification and subsequent convergence. In summary, we find that even just a few genetic differences affecting developmental and social features can greatly impact morphological evolution of multicellular bodies and experimentally demonstrate that microbial warfare can promote cooperation.

## 1. Introduction

Multicellular developmental systems involving cell differentiation, complex cell-cell interactions and collective morphological output have originated independently many times in both zygotic [1] and aggregative modes [2,3]. Once originated, developmental systems have diversified remarkably at all organizational levels from interplay among selective, historical and stochastic forces [4–6] to generate seemingly 'endless forms' [7]. Biotic selection shaping development occurs both across and within species, for example from predation [2,8] and sexual selection [9,10], respectively. Other social forces such as interference competition and cheating also have potential to shape the evolution of developmental systems but are less studied in this regard [9,11–14]. Developmental-system evolution might be shaped not only by external selective forces but also by internal system features [4–7].

Diverse prokaryotic and eukaryotic microbes exhibit aggregative multicellular development [2,15]. In some microbial systems such as the predatory myxobacteria and dictyostelids, starvation induces conspecific unicellular organisms to cooperatively develop into multicellular fruiting bodies packed with stress-resistant spores that germinate upon encountering conducive conditions [16,17]. Like zygotic developmental systems, aggregative systems have complex genetic architectures and involve temporal cascades of cell-cell signalling

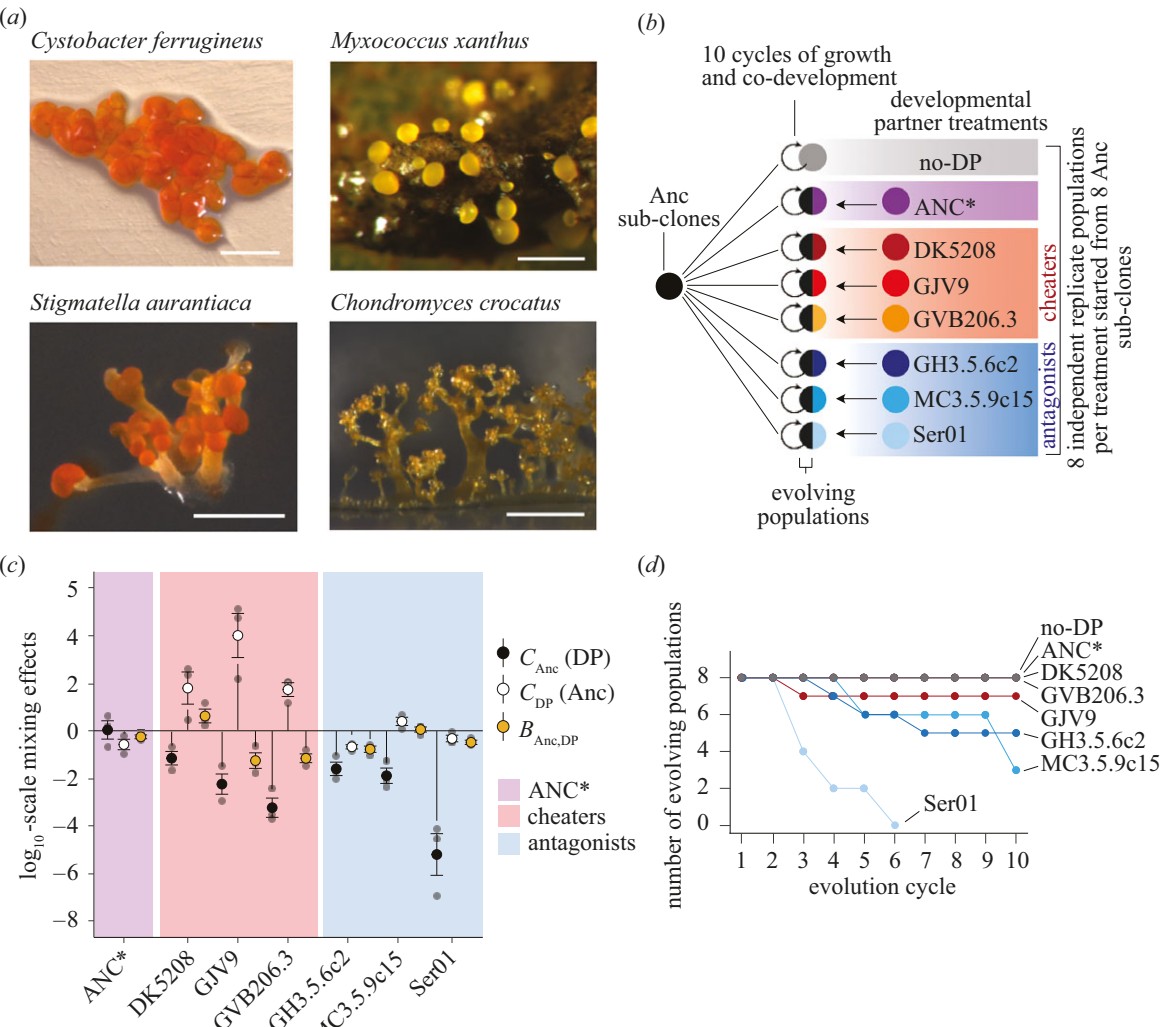

**Figure 1.** MyxoEE-7 tests whether distinct cellular environments during development shape morphological diversification in *Myxococcus xanthus*. (*a*) Fruiting body morphologies across four Myxococcales species. Left to right, top: *Cystobacter ferrugineus* (Cb f17), *M. xanthus* fruiting bodies on soil; bottom: *Stigmatella aurantiaca* (Sg a32) and *Chondromyces crocatus* (DSM 14714T). (*Myxococcus xanthus* image from Michiel Vos, others from Ronald Garcia and Rolf Müller). Scale bars are approximately 200 μm. (*b*) MyxoEE-7 design summary. Black semi-circles represent Anc-derived evolving populations that repeatedly undergo co-development with a non-evolving partner (coloured semicircles). The grey circle represents populations evolving with no developmental partner (no-DP). (*c*) Effects of mixing the MyxoEE-7 *M. xanthus* ancestor Anc 1 : 1 with each non-evolving developmental partner (DP) on spore production by Anc ($C_{Anc}$(DP), black circles), DP ($C_{DP}$(Anc), white circles) and the total group ($B_{Anc,DP}$, tan circles) relative to monoculture spore production (see Methods). Lines extending from zero indicate significant effects (one-sample *t*-tests, all *p*-values are listed in the electronic supplementary material, table S3). A zero value indicates no effect of mixing (horizontal-solid line). See the electronic supplementary material, figure S1a for pure-culture spore counts and absolute mix counts. Error bars, s.e.m. ($n = 3$). (*d*) Survival/extinction dynamics for all eight MyxoEE-7 treatments. The *y*-axis values indicate the number of independent replicate populations within each treatment defined by developmental partner (or none) at each evolution cycle.

resulting in cell-type differentiation. In the myxobacteria, fruiting body morphologies and their genetic foundations have diversified greatly [18,19] (figure 1a). Such morphological diversification may have been driven in part by variation in selective forces, whether abiotic, e.g. the physico-chemical character of soil surfaces [20], or biotic, e.g. the composition of prey communities [21] or social groups [22,23]. However, the roles of diverse potential selective forces relative to historical constraints and stochasticity in shaping extant morphological diversity are unknown.

The potential for genetic diversity among co-developing cells is an important feature of microbial aggregative development from onset to culmination. *Myxococcus* fruiting bodies emergent from natural soils are often composed of recently and locally diversified lineages and yet differentiated at the group level across small spatial scales [23,24]. Such fine-scale mosaic geographical structure of cell-group

diversity appears to persist over evolutionarily significant periods [23]. Thus, the evolution of aggregative development might be subject to group-level forms of historical contingency. The developmental evolution of a focal lineage may be contingent on not only its own genetic history (the standard conception of historical contingency [4,5]) but also on the histories of other lineages with which it interacts, co-develops, and potentially coevolves [22–25]. In the latter case, the relevant historical contingencies are also social selective forces. Even when fruiting bodies have little or no internal genetic diversity [23], the cellular environments of development generated by distinct founding genotypes might differ in their selective effects on new mutations and thus differentially shape future developmental evolution. Here, we test whether the cellular environment of microbial aggregative development matters to the morphological evolution of fruiting bodies.

Interactions during development between distinct *Myxococcus xanthus* isolates sampled from the same fruiting body are often synergistic but sometimes antagonistic [22]. Such groupmates often vary in monoculture spore production and include genotypes with severe developmental defects that can be socially complemented by developmentally proficient genotypes from the same group [22–24,26]. Some defective genotypes 'cheat' by exploiting proficient genotypes in mixed groups to gain a relative fitness advantage at spore production [27]. Soil populations are structured in a fine-scale biogeographic mosaic of antagonistic conspecifics that combat one another upon encounter and that coexist patchily, probably owing to positive frequency dependence of combat success [28]. Such conspecific warfare is common in bacteria and is expected to promote various forms of microbial cooperation [28–32].

Considering potential natural interactions among myxobacterial conspecifics differing in social character, we designed an evolution experiment named MyxoEE-7 to address questions regarding effects of cellular diversity within developmental systems on their morphological evolution. Specifically, we ask whether morphological evolution of cooperative development can be differentially shaped by: (i) differences in the cellular composition of development *per se*, (ii) major behavioural categories of developmental partners—benign cooperators, cheaters and antagonists (see Semantics in the electronic supplementary material), and (iii) small genetic differences (e.g. between distinct cheaters).

## 2. Results

Starting with eight subclones of one ancestral genotype (Anc, electronic supplementary material, table S1), we established eight replicate populations in each of eight treatments to initiate MyxoEE-7 (figure 1*b*; electronic supplementary material, figure S1*a*; Methods). Each evolutionary cycle consisted of growth in nutrient-rich liquid followed by development on starvation agar (electronic supplementary material, figure S1*a*). The cellular environment of development was manipulated in seven treatments by mixing Anc-derived populations 1 : 1 with a non-evolving developmental partner unique to each treatment at the onset of starvation (or no partner in one additional treatment) (figure 1*b*). After development, spores were heat-selected and non-evolving partners were killed with antibiotic during growth (electronic supplementary material, figure S1*a*). We thus created eight distinct treatments of replicated developmental systems. Each system had one cellular component that would contribute descendants to the next developmental cycle and another cellular component that would not, with treatments initially differing only in the identity of their non-evolving developmental partner.

We then tested whether the initially identical evolving components would morphologically diverge over time as a function of their non-evolving developmental partners. The three behavioural categories of non-evolving partners—benign cooperators, cheaters and antagonists—were hypothesized to impose different forms of social selection on Anc-derived populations during development and thus potentially differentially shape morphological evolution (electronic supplementary material, table S2). In the benign cooperator category, populations co-developed with the antibiotic-sensitive parent of the experimental ancestor Anc (ANC*, figure 1*b*,*c*; electronic supplementary material, figure S1*b* and table S1). In the cheater category, the three developmental partners were strains severely defective in monoculture sporulation that nonetheless outcompete Anc at sporulation in mixed groups (DK5208, GJV9, GVB206.3; figure 1*b*,*c*; electronic supplementary material, figure S1*b*). In the antagonist category, evolving populations co-developed with one of three genomically distant natural isolates (GH3.5.6c2, MC3.5.9c15, and Ser01) that are highly proficient at fruiting body formation and sporulation in pure culture and greatly reduce Anc spore production (by ≥ 98%) in co-developmental groups (figure 1*b*,*c*; electronic supplementary material, figure S1*b*). Cheating and antagonistic partners are diverged from Anc by small (less than 20 mutations) and large (greater than 100 000 mutations, core genome size *ca* 6.9 Mb) degrees, respectively (electronic supplementary material, table S1). In an additional treatment with no developmental partner (no-DP, figure 1*b*), the cellular composition of development was unmanipulated.

From an evo-devo perspective, the social character of aggregative development allowed us to manipulate the cellular environment of evolving developmental systems without starting evolution from multiple genetically distinct ancestors in a traditional historical-difference experiment [5]. We thus test for an effect of the cellular context of developmental systems *per se* on morphological evolution. From a social evolution perspective, we examine how repeated interaction with conspecifics of radically different social character—friendly cooperators, closely related cheaters, and distantly related antagonists—evolutionarily shapes the output of a cooperative process.

Seventeen (17) of the 64 initial MyxoEE-7 populations (27%) went extinct during the experiment, all but one of which co-developed with an antagonist ($p < 0.0001$, Fisher's exact test). All eight populations partnered with the antagonist Ser01 went extinct by the sixth cycle (figure 1*d*), an outcome explainable by the severe reductions of Anc spore production caused by co-development with Ser01 (electronic supplementary material, figure S1*b*). However, because Anc spore production was consistently approximately 1000-fold higher when mixed with GH3.5.6c2 or MC3.5.9c15 compared to mixes with Ser01 (electronic supplementary material, figure S1*b*), simple experimental variation in co-developmental outcomes is unlikely to explain the extinction events in GH3.5.9c2- and MC3.5.9c15-partnered populations. One possible explanation, among others, is the appearance of adaptive mutations that generate frequency-dependent sensitivity to antagonistic partners. In this scenario, low-frequency adaptive mutants receive social protection against antagonists from other high-frequency genotypes but increase in susceptibility to antagonists with increasing frequency. If such mutants reached sufficiently high frequency by the end of some growth phases, extinction events during subsequent development phases might result.

### (a) The cellular character of aggregative development shapes morphological evolution

Examples of population-level developmental phenotypes of one clone of the ancestor Anc and of populations derived from that clone that evolved in four different treatments at two evolutionary time points are shown in figure 2*a*. In

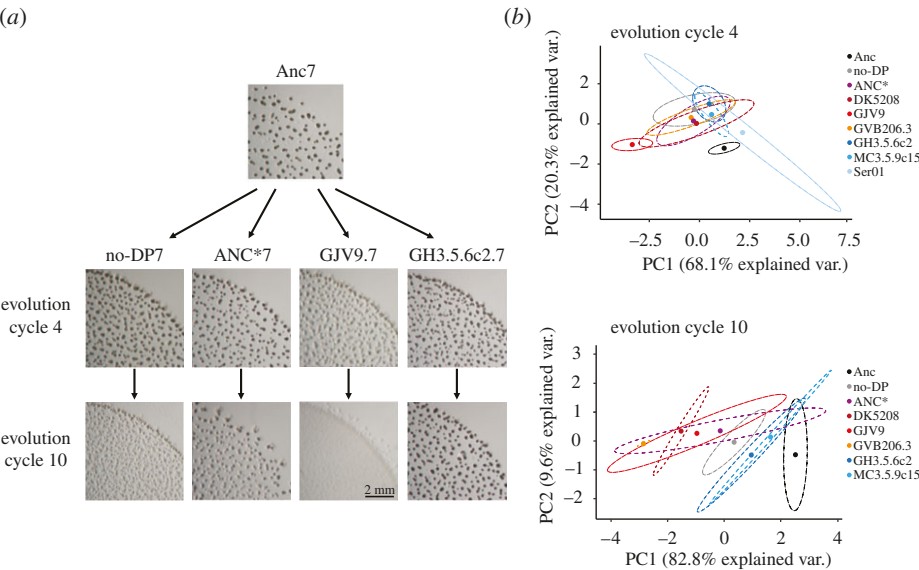

**Figure 2.** The cellular character of aggregative development shapes morphological diversification. (*a*) Developmental phenotypes of the experimental ancestor and one evolved replicate population from four of the eight developmental-partner treatments after four and 10 cycles of evolution. The evolved populations imaged here all descend from the same ancestral subclone (Anc7). (*b*) PCA-morphospace positions integrating four quantified developmental traits (electronic supplementary material, figure S2*a*) for the eight ancestral subclones (black circles and ellipses) used to initiate all evolution lines and for all treatment sets of evolved populations (coloured circles and ellipses) after four and 10 cycles of evolution. Circles represent PCA-morphospace centroids obtained from three biological-replicate assays surrounded by 95% confidence regions (dashed ellipses) (*n* = 3; electronic supplementary material, table S4). Percentage values on the *x*- and *y*-axes indicate the proportion of variation (var.) explained by the first and second principal components (PC1 and PC2), respectively (electronic supplementary material, figure S2*b*).

comparison with the ancestor Anc, we quantified four morphological traits (see Methods) of all surviving populations in the absence of their developmental partner (if any) after four and 10 cycles of experimental evolution (see Methods), thereby characterizing how intrinsic developmental phenotypes evolved in response to co-development with the various non-evolving partners. One trait—the number of fruiting bodies per plate—is a population-level phenotype whereas the three other traits are features of individual fruiting bodies—area, density and density-heterogeneity (see Methods). We first integrate all traits in multivariate principal component analysis (PCA) to characterize overall morphological evolution and subsequently examine evolutionary patterns of individual traits.

The overall morphological analysis demonstrates that: (i) treatment-level sets of derived populations diverged from their ancestor, and (ii) the different non-evolving developmental partners caused their corresponding replicate-population sets to diversify at both the individual-treatment level and the treatment-category level (figure 2*b*; electronic supplementary material, figure S2 and table S4). For example, already by cycle 4 most evolved treatments had diverged morphologically from Anc and the GJV9-partnered cheater treatment had collectively diverged from most other treatments (figure 2*b*). By cycle 10, overall treatment-level divergence across the morphospace increased further still, with all three cheater-partner treatments showing morphological divergence from the two surviving antagonist-partner treatments (figure 2*b*). Evidence of treatment divergence is additionally supported by hierarchical clustering analysis (see the electronic supplementary material) of overall morphological similarity, which indicates the evolution of three distinct morphological treatment clusters by cycle 10 (electronic supplementary material, figure S2*c*). Each cluster is composed of a pair of treatments that are morphologically more similar to one another than to any other treatment—a

pair of cheater treatments (DK5208 and GJV9), the pair of surviving antagonist treatments and a pair formed by the benign cooperator ANC* treatment with the no-DP treatment (electronic supplementary material, figure S2*c*). Importantly, the morphological divergence observed at the treatment level demonstrates most fundamentally that populations underwent adaptation during the experiment and that some adaptation was specific to interaction with particular developmental partners. Because only the cellular environment of development differed systematically across treatments, only differential adaptation to distinct developmental partners can plausibly explain treatment-level divergence at heritable traits. The cellular character of aggregative developmental systems shaped the course of morphological evolution.

Except for fruiting body counts at cycle 4, average trait values across the entire experiment tended to decrease during evolution (figure 3*a*), but this general trend masks rich variation in dynamics across, and in some cases within, developmental-partner categories. Consistent with the divergent position of GJV9-partnered populations in PCA morphospace (figure 2*b*), this cheater treatment showed unique evolutionary patterns at all four analysed traits by cycle 4, decreasing rather than increasing in fruiting body number and decreasing more than any other treatment at the other three traits (electronic supplementary material, figure S3*a*). Later, ANC* and the cheaters DK5208 and GVB206.3 drove large treatment-level decreases in fruiting body number to levels near or below that of the GJV9-partnered treatment at cycle 10, which decreased only slightly, if at all, in the later cycles (electronic supplementary material, figure S3*a*). In one particularly striking comparison, GVB206.3- versus GJV9-partnered populations diverged greatly in fruiting body number by cycle 4, with the GVB206.3 treatment making far more fruiting bodies, but then strongly reversed rank during later evolution (electronic

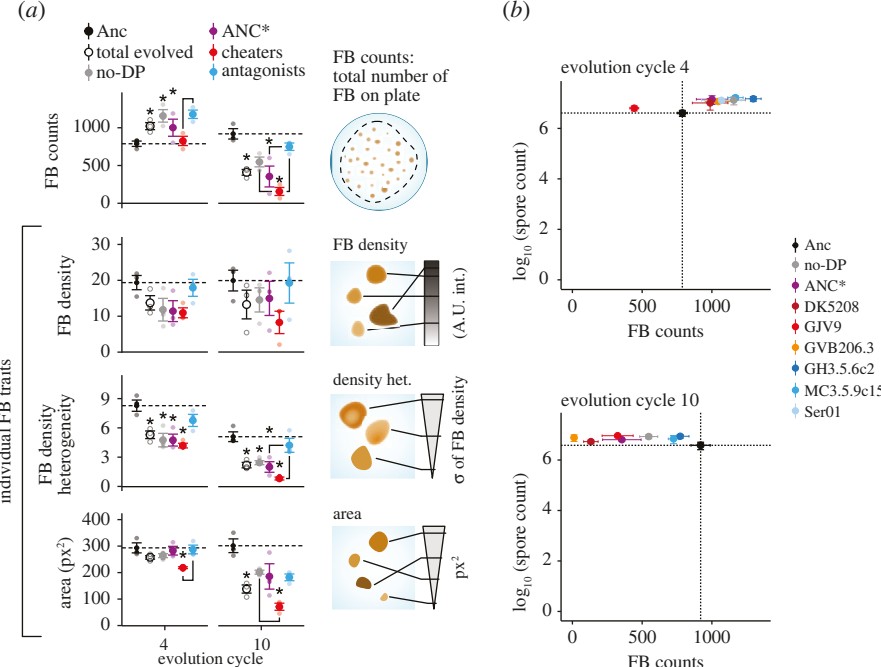

**Figure 3.** Social antagonists promote evolutionary maintenance of cooperative aggregative development. (*a*) Phenotype means of the eight ancestral sub-clones (black circles), all evolved populations considered together (open circles), populations evolved with no partner (no-DP, grey circles), evolved populations partnered with ANC* (purple circles), all surviving populations partnered with a cheater (red circles) and all surviving populations partnered with an antagonist (blue circles) after four and 10 evolutionary cycles. Circles represent cross-assay means (*n* = 3) of within-assay trait averages across all populations within each category. Stars represent significant differences relative to Anc and line connectors indicate significant differences between the respective categories of evolved populations (two-tailed *t*-tests were used to compare all evolved populations together (total evolved) versus Anc, one-way ANOVA and subsequent two-tailed Tukey tests were used for all other contrasts, *p*-values reported in the electronic supplementary material, tables S5–S7). (*b*) Ancestral and treatment-level evolved spore production versus fruiting body (FB) counts. In both (*a*) and (*b*) ancestor means can differ between cycle 4 and cycle 10 plots because they were obtained independently for the two sets of separately-performed assays. In both (*a*) and (*b*), error bars represent s.e.m.

supplementary material, figure S3*a*). The GVB206.3-partnered populations dropped to near zero fruiting bodies at cycle 10 while the GJV9-partnered populations continued to form hundreds of fruiting bodies on average (electronic supplementary material, figure S3*a*). Thus, one cheater (GJV9) drove a much more rapid decrease of fruiting body formation while another (GVB206.3) ultimately drove a more complete loss. Given that ANC* and the three cheater partners are known or expected to differ from one another by only approximately 7–30 mutations (electronic supplementary material, table S1), the unique trajectory of GJV9-partnered populations demonstrates that even small historical differences between developmental partners of focal evolving lineages can greatly impact morphological evolution of the latter. In social terminology, even small historical differences between group-mates sharing a major behavioural characteristic—in this case cheating—can greatly impact how social phenotypes of focal lineages evolve.

## (b) Distinct social interactant categories differentially affect morphological evolution

Morphological evolution varied not only between some individual treatments, but also among the three major partner categories defined by behavioural phenotypes—the benign parent (ANC*) of the experimental ancestor, cheaters, and antagonists (figure 3*a*; electronic supplementary material, table S2). By cycle 4, surviving populations that co-developed with an antagonist collectively evolved higher fruiting body counts than cheater-partnered populations (figure 3*a*). By cycle 10, all categories decreased in fruiting body counts

from their respective cycle 4 values, but to different degrees. Specifically, ANC*- and cheater-partnered populations dropped to fruiting body counts far below those of the Anc sub-clones, whereas antagonist-partnered populations continued to produce nearly as many fruiting bodies as Anc (figure 3*a*). After 10 cycles, ANC*- and cheater-partnered populations formed only 47% and 20% as many fruiting bodies as antagonist-partnered populations, respectively (figure 3*a*). These large relative losses in fruiting body numbers were not countered by relative gains at other traits indicative of developmental proficiency, particularly fruiting body density or area. Indeed, fruiting body-density and fruiting body-area means for antagonist-partnered populations were either greater than or indistinguishable from means for the other categories at both evolved time points (figure 3*a*; electronic supplementary material, figure S3*a*). These patterns collectively show that developmental interaction with antagonists selectively favoured maintenance of robust fruiting body formation (e.g. figure 2*a*) and thereby evolutionarily stabilized developmental cooperation.

The magnitude of morphological differentiation between treatments was not found to correlate with genomic divergence between respective developmental partners, as might be expected [5] (electronic supplementary material, table S1). Small genomic differences between GJV9 and the two other cheating partners (approx. 30 or fewer mutations, electronic supplementary material, table S1) caused greater divergence between their respective treatments at cycle 4 than did the much larger genomic difference (electronic supplementary material, table S1) between the antagonists GH3.5.6c2 and MC3.5.9c15 (figure 2*b*; electronic supplementary material, figures S2*c* and S3*a*). Collectively, these

outcomes indicate that small historical differences between developmental partners can strongly shape divergence of focal interactant lineages and suggest that some behavioural characteristics, e.g. antagonism, shared by even highly divergent partners can matter more for shaping developmental evolution than do large genomic differences.

## (c) Sporulation is evolutionarily independent from fruiting body formation

Previous work suggested that evolutionary decreases in the ability to form many dense fruiting bodies in pure culture would generally be associated with large decreases in spore production [33], but this was not the case here. In all treatments except the GJV9-partnered treatment, both monoculture spore production and monoculture fruiting body counts remained at or increased over the ancestral level by cycle 4 (figure 3b; electronic supplementary material, figure S3b). However, among the GJV9-partnered populations at cycle 4, sporulation remained high while fruiting body counts decreased greatly (approx. 40%) (figure 3b; electronic supplementary material, figure S3), a pattern revealing evolutionary independence of prolific starvation-induced spore production from fruiting body number. While *M. xanthus* can undergo a form of individualistic, chemical-induced sporulation by different pathways than starvation-induced sporulation [34,35], pervasive evolutionary uncoupling of high levels of starvation-induced spore production from fruiting body formation was previously unknown. Further illustrating this independence, monoculture spore production remained at or above ancestral levels in all treatments at cycle 10 (electronic supplementary material, figure S3b) despite large decreases in fruiting body counts, fruiting body density and area in most treatments (figure 3; electronic supplementary material, figure S3). Most strikingly, sporulation and fruiting body formation became completely uncoupled among the GVB206.3-partnered populations, which made almost no fruiting bodies at cycle 10 while retaining ancestral levels of monoculture sporulation. The observation that fruiting body formation is not required for extensive starvation-induced sporulation highlights that the fundamental question of exactly why, from an evolutionary perspective, myxobacteria form fruiting bodies remains unclear, although several hypotheses have been proposed [36,37].

## (d) Social selection limits the degree of stochastic morphological diversification

We additionally tested whether differences in adaptive-landscape topology imposed by distinct developmental partners differentially constrained morphological diversification among replicate populations within treatments [38,39] (see the electronic supplementary material)—diversification which is caused by stochastic variation in mutational input. After four cycles, cheater-partnered population sets collectively evolved greater degrees of inter-population diversity within treatments than was present among the Anc sub-clones but antagonist-partnered populations did not (figure 4). However, differential patterns of stochastic diversification were dynamic, shifting and even reversing over later cycles. Replicate populations with no partner and those partnered with ANC* began and continued increasingly diversifying by cycles 4 and 10, respectively. By contrast,

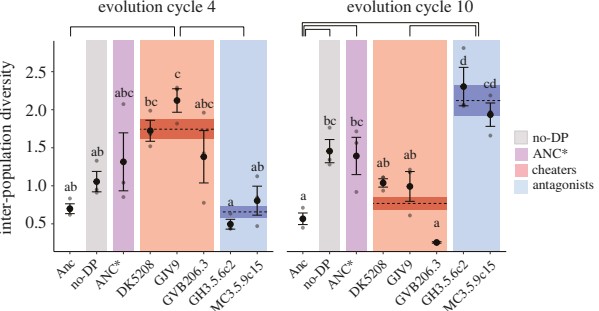

**Figure 4.** Developmental partners differentially determine patterns of stochastic morphological diversification and deterministic convergence. Quantification of morphological diversification (see the electronic supplementary material) among the eight Anc subclones and among evolved populations within each cellular-environment treatment and treatment category after four and 10 evolution cycles. Coloured shading highlights treatment categories. Dashed lines indicate treatment-category means and surrounding darker shading indicates the corresponding s.e.m. Non-overlapping letter sets and connecting solid-black lines indicate significant differences between mean values of treatment and treatment categories, respectively (one-way ANOVA followed by two-tailed Tukey tests; electronic supplementary material, tables S8–10). The Ser01-antagonist treatment was not analysed owing to many early extinction events. Error bars, s.e.m. ($n = 3$). Note that categories belonging to the same statistical class does not imply morphological similarity between the categories but only indistinguishable degrees of within-category diversification. Compared categories might have identical diversification means while having very different trait means.

antagonist-partner treatments underwent highly parallel evolution in the early cycles, only to later undergo a relative explosion of morphological diversification. In particular, the GH3.5.6c2 treatment ended MyxoEE-7 more diversified than all non-antagonist treatments. In yet further contrast, during early evolution the populations within cheater-partnered treatments underwent the greatest diversification but then later converged towards morphological similarity (figure 4) while continuing to collectively diverge from their ancestor in trait means (figure 3a). Strikingly, GVB206.3-partnered populations converged to near morphological identity (figure 4). Late convergence within the cheater-partner treatments reflects trajectories of greatly decreased fruiting body numbers, density and area (figure 3; electronic supplementary material, figure S3). In summary, cellular environments differentially determine the temporal dynamics and magnitude of stochastic developmental diversification and post-diversification convergence.

## 3. Discussion

Social forces such as cooperation, cheating and interference competition have the potential to influence developmental processes and their evolutionary trajectories [9,13,14]. In aggregative developmental systems, such social forces occur within the developmental process because conspecifics are the cells that carry out development. In this study, we manipulated the cellular composition of aggregative multicellular development in the bacterium *M. xanthus* during MyxoEE-7 (figure 1b; electronic supplementary material, figure S1a; see Methods) and demonstrated that the character of cellular interactants within developmental systems profoundly shapes the

future morphological evolution of those systems. Distinct developmental partners strongly influenced both the deterministic evolution (figures 2 and 3) and stochastic diversification (figure 4) of developmental phenotypes. Moreover, such developmental-partner effects were observed at multiple levels of categorization, both between treatments in which distinct cheating developmental partners differed by only several mutations and between treatment sets that collectively differed in a major behavioural feature of the developmental partner, e.g. cheating versus antagonism.

Biotic antagonisms strongly shape the evolution of developmental and social systems. Across species, predation drives much morphological evolution in prey [40,41], selects for cooperative prey defence [42] and can promote the evolution of multicellularity [8,43]. Within species, interference competition shapes animal cooperation [44] and is proposed to evolutionarily promote some forms of microbial cooperation, whether directly or indirectly [28,29,31,32,45]. The uniquely high retention of the intrinsic ability to form robust fruiting bodies among the antagonist-partnered populations of MyxoEE-7 experimentally demonstrates that interaction with hostile conspecifics can directly favour a microbial cooperative trait. Mechanistically, antagonists may have disfavoured mutations decreasing costly production of extracellular adhesins necessary for robust fruiting body formation [46], mutations that may have been advantageous in non-antagonist treatments. Alternatively, antagonists may have selected for mutations altering motility behaviour that generate preferential aggregation among Anc-derived cells into fruiting bodies with fewer antagonist cells.

Our results suggest that conspecific interference competition may, in concert with other forces (electronic supplementary material, table S2), promote maintenance of fruiting body formation in the wild and may have even contributed to its original emergence. In *Myxococcus*, success at contact-dependent warfare is positively frequency dependent [28]. If a geographical mosaic of antagonist-type territories [22,28,31,32,47] existed prior to the origin of fruiting bodies, evolving the ability to aggregate and cohere prior to sporulation may have generated benefits of larger dispersal groups for success at conspecific combat upon dispersal.

More broadly, effects of conspecific interference competition on the evolution of microbial cooperation and multicellularity require further investigation. Such effects might be direct responses to antagonism-imposed selection—as occurred in this study—or indirect by-products resulting from toxin-based warfare contributing to high-relatedness population structures conducive to the evolution of cooperative traits [28,31,32]. Evolution experiments can be conducted to test whether and how different mechanistic forms of warfare promote microbial behaviours involving potentially exploitable secretions, including biofilm formation and counter-weaponry. With both microbial and larger organisms, comparative effects on social evolution of interaction with antagonistic, cheating and benignly cooperative conspecifics, singly or in combination, can be examined in both monospecific and community contexts.

Our finding that distinct developmental partners can greatly impact morphological evolution in *M. xanthus* may help to understand the remarkable diversification of fruiting body morphologies observed across myxobacterial species (figure 1*a*). Distinct cellular compositions of spatially clustered lineage sets during natural cycles of development may have contributed significantly to this diversification by differentially impacting evolutionary trajectories [18,19]. Across a broader range of multicellular developmental systems, the cellular character of development may shape not only both deterministic and stochastic forms of morphological evolution, but also the evolution of phenotypic plasticity [48–50], developmental bias [51] and both the genetic basis [52] and evolvability of development [53,54].

## 4. Methods

### (a) Experimental evolution: MyxoEE-7
Eight subclones of GJV27 (here aka 'Anc') were isolated (GJV27.1/ Anc1 - GJV27.8/Anc8), stored frozen and used to establish eight replicate populations for each of eight evolutionary treatments (64 populations total, figure 1*b*). Anc subclones and the seven non-evolving partner strains were initially grown separately in CTT [55] liquid (32°C, 300 r.p.m.) (electronic supplementary material, figure S1*b*) from frozen stocks. Mid-log-phase cultures were centrifuged (5 min at 12 000 r.p.m.) and resuspended to approximately $5 \times 10^9$ cells ml$^{-1}$ in TPM liquid buffer [55]. Anc-subclone populations were mixed with the seven partners in a 1:1 ratio and 50 µl of the mixes (approx. $2.5 \times 10^8$ cells total, approx. $1.25 \times 10^8$ cells of each partner) were spotted on TPM starvation-agar plates (1.5% agar) [55]. A 50 µl aliquot of each resuspended Anc-subclone culture was also spotted without mixing for the control treatment (no-DP) lacking a non-evolving partner.

Populations were incubated on TPM agar for 5 days (32°C, 90% relative humidity (RH)) and heated at 50°C for 3 h to kill non-spores. Cells were harvested by scalpel and transferred into CTT liquid with kanamycin, which killed the non-evolving partners (electronic supplementary material, figure S1*c*). Cultures of the Anc-derived populations were then grown (32°C, 300 r.p.m.) to mid-log phase. Cultures were deemed extinct if no growth was evident after 5 days. To eliminate residual kanamycin prior to mixing Anc-derived cultures with their respective non-evolving partner for the next developmental phase, evolving cultures were centrifuged as above, washed by resuspension in 2 ml TPM liquid and centrifuged again prior to final resuspension at approximately $5 \times 10^9$ cells ml$^{-1}$ in TPM liquid. For each subsequent cycle, cultures of non-evolving partners were initiated from the same frozen stock, grown in CTT liquid, and processed as in the first cycle. Mixes of evolving populations with non-evolving partners and cultures of the no-DP populations were then processed as in the first cycle to initiate subsequent starvation cycles (TPM agar, 5 days (occasionally 4)). Samples of evolving populations were stored frozen (−80°C, 20% glycerol) at the end of each growth phase, including a growth phase following cycle 10. See the electronic supplementary material for estimates of population bottlenecks during the evolution experiment. After five cycles, all populations were discarded owing to media contamination and subsequently re-initiated from stocks frozen after cycle 4.

We name this evolution experiment 'MyxoEE-7', with 'MyxoEE' meaning 'Myxobacteria Evolution Experiment' and '7' indicating the temporal rank position of this first publication from MyxoEE-7 relative to the first publications from other MyxoEEs [56]. Prior MyxoEEs from which studies have been published are correspondingly named MyxoEE-1 [33], MyxoEE-2 [57,58], MyxoEE-3 [25,56,59–62], MyxoEE-4 [63], MyxoEE-5 [64] and MyxoEE-6 [21].

### (b) Development assays
Frozen-stock samples of the ancestral subclones Anc1-Anc8 and all evolved populations from cycle 4 or 10 were inoculated and grown

in CTT liquid with kanamycin to mid-exponential phase. We subsequently centrifuged (5 min at 12 000 r.p.m.) and resuspended cells in TPM liquid twice to a final density of approximately $5 \times 10^9$ cells ml$^{-1}$. We spotted 50 µl of the resuspension on plates (6 cm diameter) containing 5 ml of TPM hard (1.5%) agar, allowing the resulting spots to dry for 1 h in a laminar-flow hood. Once dried, plates were incubated upside down at 32°C, 90% RH for 5 days before microscopy images were taken. Developmental assays were replicated three times independently at separate times. Assays of cycle 4 and cycle 10 populations were performed separately, each including all Anc subclones in every replicate. Morphological assays were performed separately from sporulation assays.

## (c) Imaging and trait quantification

Images of evolved populations shown in figure 2a were obtained with a Zeiss STEMI 2000 microscope and a Nikon Coolpix S10 camera. Images for quantitative morphological analysis were taken using an Olympus SXZ16 microscope with an Olympus DP80 camera system. The image-acquisition protocol was identical across samples (exposure time = 9.9 ms, lens = Olympus 0.5 × PF, zoom = 1.25x, ISO = 200, illumination = BF built-in system). Images were processed with Fiji software [65] by duplication, conversion to 8-bit mode and segmented using the Triangle algorithm to identify individual fruiting bodies. Dust particles that appeared while imaging were masked before segmentation. Segmentation was performed over the image area covered by the spotted cell population. Segmented objects with an area value smaller than 20 px$^2$ were excluded while all larger objects were retained for analysis. No maximum size limit or circularity restrictions were imposed. Regions obtained in this manner were over-imposed on the original images and used to define the area in which phenotypes were quantified.

After fruiting body definition, four traits were used to characterize developmental morphology:

(i) *fruiting body number* [20,66,67]: total number of fruiting bodies generated by approximately $2.5 \times 10^8$ initial cells per plate;
(ii) *fruiting body density*: average grey-value intensity of pixels per fruiting body. This measurement is expressed in arbitrary-unit grey-intensity values, which are expected to correlate with total cell/spore density within fruiting bodies. This trait has been previously used in referring to 'mature' fruiting bodies [20];

(iii) *density heterogeneity*: standard deviation of within-fruiting body pixel-grey values (fruiting body density). This parameter quantifies the heterogeneity of cell density within a developmental aggregate and it has been studied with different methods in earlier works [68,69]; and
(iv) *fruiting body area* [20,66,67,70]: surface area occupied by each fruiting body expressed in total pixel number within the defined fruiting body border.

For further analysis, we used median values of fruiting body density, density heterogeneity and area measurements per plate. Trait values for all eight Anc subclones were obtained in parallel with evolved populations during each assay replicate. For most analyses, trait values were averaged across replicate populations from the same evolutionary treatment (or across Anc subclones) for each replicate assay and the resulting treatment-level means were subsequently averaged across three biological replicates.

## (d) Spore counts and mixing effect quantification

Spore counts were obtained as in [71], except that 50°C heat selection was imposed for 3 h prior to harvest. Effects of mixing Anc and each non-evolving partner reported in figure 1b on each strain individually ($C_i(j)$) and total mixed-group productivity ($B_{ij}$) were also calculated as in [71].

## (e) Statistical analyses

See the electronic supplementary material.

Data accessibility. All raw data and representative code are available from the online repository FigShare: https://doi.org/10.6084/m9.figshare.14905038 [72].

Authors' contributions. M.L.F.: conceptualization, data curation, formal analysis, funding acquisition, investigation, methodology, validation, writing—original draft, writing—review and editing; G.J.V.: conceptualization, funding acquisition, project administration, resources, supervision, writing—original draft, writing—review and editing. Both authors gave final approval for publication and agreed to be held accountable for the work performed therein.

Competing interests. We declare we have no competing interests.

Funding. This work was funded in part by an EMBO Long-Term Fellowship (ALTF 1208–2017) to M.L.F. and SNSF grant no. 31003B_6005 to G.J.V.

Acknowledgments. The authors thank Ronald Garcia and Rolf Müller for providing images of myxobacterial fruiting bodies and Samay Pande, Marie Vasse, Mary Jane West-Eberhard and Sébastien Wielgoss for comments on the manuscript.

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
