## [Peer Review File · Proceedings of the Royal Society B: Biological Sciences]

Review History

RSPB-2021-1522.R0 (Original submission)

Review form: Reviewer 1

Recommendation

Accept with minor revision (please list in comments)

Scientific importance: Is the manuscript an original and important contribution to its field?

Good

General interest: Is the paper of sufficient general interest?

Good

Quality of the paper: Is the overall quality of the paper suitable?

Good

Is the length of the paper justified?

Yes

Should the paper be seen by a specialist statistical reviewer?

No

Do you have any concerns about statistical analyses in this paper? If so, please specify them explicitly in your report.

No

It is a condition of publication that authors make their supporting data, code and materials available - either as supplementary material or hosted in an external repository. Please rate, if applicable, the supporting data on the following criteria.

Is it accessible?

Yes

Is it clear?

Yes

Is it adequate?

Yes

Do you have any ethical concerns with this paper?

No

Comments to the Author

This paper is interesting. First off, it's the only paper I've ever seen that does experimental evolution with no social partners, antagonists, and social cheats. That's a cool premise! The data / writing were dense and took a bit of thinking to figure out. But I think there's some interesting stuff in here.

Specifically, I thought it was surprising and quite cool that antagonists maintained multicellular group formation more effectively than cheats or even a lack of social partners. The explanation given, in which antagonists may drive high spatial assortment prior to initiation of fruiting body formation, is compelling (though not directly tested, but I think that could be work for a future paper).

I think this work provides interesting insight into how microbial interactions affect the evolution of aggregative multicellularity, which is an important and understudied topic in the evolution of multicellularity. My only real suggestion for improvement is to explain some of the features of the system for those of us that are not steeped in myxo. The overall rapid loss of fruiting bodies in most lineages (even the no social partner)- is that due to the decoupling of sporulation from fruiting body formation? If so, can you expand upon that a bit in the discussion? What is the significance of fruiting bodies in myxo's ecology, and do you typically see rapid loss of fruiting bodies with lab evolution experiments? Is the fact that they are maintained when being grown with antagonists simply an artifact of increased spatial assortment, or is there a selective advantage to maintaining fruiting body formation when growing with an antagonist? Is it possible that in the cheating treatment, fruiting body formation was actually costly relative to unicellular sporulation, given the social exploitation? Clearly you do not need to answer all these questions, these are just what arose when thinking about your work. I think a more extensive discussion explaining how your results impact our thinking about the evolution of multicellularity / microbial social evolution would be helpful.

Anyway, it's cool work and a neat experimental system that I expect will continue to be insightful.

Review form: Reviewer 2

Recommendation

Accept with minor revision (please list in comments)

Scientific importance: Is the manuscript an original and important contribution to its field?

Good

General interest: Is the paper of sufficient general interest?

Good

Quality of the paper: Is the overall quality of the paper suitable?

Acceptable

Is the length of the paper justified?

Yes

Should the paper be seen by a specialist statistical reviewer?

Yes

Do you have any concerns about statistical analyses in this paper? If so, please specify them explicitly in your report.

Yes

It is a condition of publication that authors make their supporting data, code and materials available - either as supplementary material or hosted in an external repository. Please rate, if applicable, the supporting data on the following criteria.

Is it accessible?

N/A

Is it clear?

N/A

Is it adequate?

N/A

Do you have any ethical concerns with this paper?

No

Comments to the Author

I really enjoyed reading your manuscript. there are few recommendations I give you. I hope other works focus more on these topics

Decision letter (RSPB-2021-1522.R0)

07-Oct-2021

Dear Dr La Fortezza

I am pleased to inform you that your Review manuscript RSPB-2021-1522 entitled "Social selection within aggregative multicellular development drives morphological evolution" has been accepted for publication in Proceedings B.

The referee(s) do not recommend any further changes. Therefore, please proof-read your manuscript carefully and upload your final files for publication. Because the schedule for publication is very tight, it is a condition of publication that you submit the revised version of your manuscript within 7 days. If you do not think you will be able to meet this date please let me know immediately.

To upload your manuscript, log into <http://mc.manuscriptcentral.com/prsb> and enter your Author Centre, where you will find your manuscript title listed under "Manuscripts with Decisions." Under "Actions," click on "Create a Revision." Your manuscript number has been appended to denote a revision.

You will be unable to make your revisions on the originally submitted version of the manuscript. Instead, upload a new version through your Author Centre.

1) A text file of the manuscript (doc, txt, rtf or tex), including the references, tables (including captions) and figure captions. Please remove any tracked changes from the text before submission. PDF files are not an accepted format for the "Main Document".

2) A separate electronic file of each figure (tiff, EPS or print-quality PDF preferred). The format should be produced directly from original creation package, or original software format. Please note that PowerPoint files are not accepted.

3) Electronic supplementary material: this should be contained in a separate file from the main text and the file name should contain the author's name and journal name, e.g. `authorname_procb_ESM_figures.pdf`

All supplementary materials accompanying an accepted article will be treated as in their final form. They will be published alongside the paper on the journal website and posted on the online figshare repository. Files on figshare will be made available approximately one week before the accompanying article so that the supplementary material can be attributed a unique DOI. Please see: <https://royalsociety.org/journals/authors/author-guidelines/>

4) Data-Sharing and data citation

It is a condition of publication that data supporting your paper are made available. Data should be made available either in the electronic supplementary material or through an appropriate repository. Details of how to access data should be included in your paper. Please see <https://royalsociety.org/journals/ethics-policies/data-sharing-mining/> for more details.

<http://datadryad.org/submit?journalID=RSPB&manu=RSPB-2021-1522> which will take you to your unique entry in the Dryad repository.

Once again, thank you for submitting your manuscript to Proceedings B and I look forward to receiving your final version. If you have any questions at all, please do not hesitate to get in touch.

Sincerely,
Dr Daniel Costa
mailto:proceedingsb@royalsociety.org

Reviewer(s)' Comments to Author:

Referee: 1

Comments to the Author(s)

This paper is interesting. First off, it's the only paper I've ever seen that does experimental evolution with no social partners, antagonists, and social cheats. That's a cool premise! The data / writing were dense and took a bit of thinking to figure out. But I think there's some interesting stuff in here.

Specifically, I thought it was surprising and quite cool that antagonists maintained multicellular group formation more effectively than cheats or even a lack of social partners. The explanation given, in which antagonists may drive high spatial assortment prior to initiation of fruiting body formation, is compelling (though not directly tested, but I think that could be work for a future paper).

I think this work provides interesting insight into how microbial interactions affect the evolution of aggregative multicellularity, which is an important and understudied topic in the evolution of multicellularity. My only real suggestion for improvement is to explain some of the features of the system for those of us that are not steeped in myxo. The overall rapid loss of fruiting bodies in most lineages (even the no social partner)- is that due to the decoupling of sporulation from fruiting body formation? If so, can you expand upon that a bit in the discussion? What is the significance of fruiting bodies in myxo's ecology, and do you typically see rapid loss of fruiting bodies with lab evolution experiments? Is the fact that they are maintained when being grown with antagonists simply an artifact of increased spatial assortment, or is there a selective advantage to maintaining fruiting body formation when growing with an antagonist? Is it possible that in the cheating treatment, fruiting body formation was actually costly relative to unicellular sporulation, given the social exploitation? Clearly you do not need to answer all these questions, these are just what arose when thinking about your work. I think a more extensive discussion explaining how your results impact our thinking about the evolution of multicellularity / microbial social evolution would be helpful.

Anyway, it's cool work and a neat experimental system that I expect will continue to be insightful.

Referee: 2

Comments to the Author(s)

I really enjoyed reading your manuscript. there are few recommendations I give you. I hope other works focus more on these topics

Decision letter (RSPB-2021-1522.R1)

02-Nov-2021

Dear Dr La Fortezza

I am pleased to inform you that your manuscript entitled "Social selection within aggregative multicellular development drives morphological evolution" has been accepted for publication in Proceedings B.

Data Accessibility section

Open Access

Paper charges

Sincerely,

Proceedings B
